# Changes in Synthetic Soda Ash Production and Its Consequences for the Environment

**DOI:** 10.3390/ma15144828

**Published:** 2022-07-11

**Authors:** Marcin Cichosz, Urszula Kiełkowska, Kazimierz Skowron, Łukasz Kiedzik, Sławomir Łazarski, Marian Szkudlarek, Beata Kowalska, Damian Żurawski

**Affiliations:** 1Department of Chemical Technology, Faculty of Chemistry, Nicolaus Copernicus University in Toruń, 7 Gagarin Street, 87-100 Toruń, Poland; ulak@umk.pl; 2CIECH R&D Sp. z o.o., 62 Wspólna Street, 00-684 Warszawa, Poland; kazimierz.skowron@ciechgroup.com (K.S.); lukasz.kiedzik@ciechgroup.com (Ł.K.); marian.szkudlarek@ciechgroup.com (M.S.); beata.kowalska@ciechgroup.com (B.K.); damian.zorawski@ciechgroup.com (D.Ż.); 3MCMP Sp. z o.o., 5 Świerkowa Street, 86-300 Grudziądz, Poland; lazarski.mcmp@gmail.com; 4Faculty of Civil Engineering and Environmental Sciences, Bialystok University of Technology, 45A Wiejska Street, 15-351 Białystok, Poland

**Keywords:** soda ash, carbonization, non-equilibrium, NaHCO_3_, ammoniated brine

## Abstract

This publication presents a series of data of one of the most difficult chemical processes to implement in industrial conditions. Obtaining soda using the Solvay technique is a process with a world volume of about 28 Tg per year. The process is extremely physico-chemically complex and environmentally burdensome. The paper presents information on a multi-component system containing three phases with a chemical reaction. Calculations for such systems and their engineering are very complicated, but the authors show how the results of this work can be applied. This paper also describes modifications of the soda process to minimize the environmental burden and minimize the production input of Na_2_CO_3_. The modifications were beneficial in reducing CO_2_ emissions and increased the efficiency of the soda process, resulting in a measurable financial benefit. At the scale of the plant where the experiment was carried out, this reduction in CO_2_ emissions amounts to 7.93 Gg per year.

## 1. Introduction

Human influence has warmed the climate at a rate that is unprecedented for at least the last 2000 years [1]. Soda ash is produced by two main methods: “natural”, involving tron extraction and processing, and “synthetic”, a Solvay process with several modifications. Niche productions involve very small quantities compared to the Solvay technique. The state-of-the-art for synthetic soda production was established in [2]. The synthetic soda ash industry, as implemented worldwide, is one of the most environmentally burdensome inorganic industries. Waste streams in the form of CO_2_ emissions and waste suspension have not found effective management to date. These two streams are the main source of synthetic soda ash industry by-products. Various solutions have been presented in publications, but few have been implemented in industrial plants [3,4,5,6,7]. Kasikowski et al., in their work [8], presented a reduction in the negative impact of a synthetic (based on the Solvay process) soda ash installation on the natural environment. This consisted of the use of a desulfurization process. For this purpose, an intermediate from the technological process of soda production was used. Similar solutions to the use of solid waste in the Solvay process are proposed by Steinhauser [6]. Soda ash is mainly used in the glass and chemical industries, in detergents and soaps, and by individual consumers. The global soda ash market is projected to reach USD 22 billion [9]; the synthetic soda ash industry accounts for about half of this market value. Industrial owners and shareholders are reluctant to invest in additional technological operations to achieve an environmental impact, so an interesting solution was presented by Foster in his article [10] that presents a series of modifications to the process of carbon dioxide adsorption in ammonia brine to reduce CO_2_ emissions, by increasing the efficiency of the soda process. The aim of the conducted works was, above all, to increase the sodium efficiency of the process. The consequence of these changes is a reduction in CO_2_ emissions and improvement in the temperature conditions of the carbonization process.

Equilibrium studies conducted on systems of CO_2_-H_2_O-NaCl, NH_3_-CO_2_-H_2_O, and CO_2_-H_2_O-NH_3_-NaCl are of great importance in an environmental context. On the one hand, such systems serve to capture and store CO_2_ and, on the other hand, by changing the balance in these systems it is possible to significantly reduce the CO_2_ emitted into the atmosphere, thereby reducing their carbon footprint.

### 1.1. Manufacture of Additionally Ammoniated Brine

To conduct research work related to dosing additionally ammoniated brine (*AAB*) into a carbonization column, this brine must be obtained. *AAB* can be obtained from two process streams, namely, ammonia brine and pre-carbonated ammonia brine. While realizing the ammonization of ammoniated brine (*AB*) and pre-carbonated ammoniated brine (*PC-AB*) using the *MAB* plant, a number of technical, technological and analytical experiments were carried out. A strictly defined stream of ammoniated brine was obtained with a direct alkalinity of 120–160 mmol∙20 cm^−1^, i.e., approx. 6.00–8.00 mol∙dm^−3^, which was the basic assumption of the conducted research. The aim of the ammonia absorption process was to obtain a stable stream of *AAB*, characterized by a direct alkalinity of about 130 mmol∙20 cm^−3^, and to determine the concentration of chloride ions and CO_2_ before and after the absorption process. During the work, favorable conditions were also created to conduct research on the higher concentration of ammonia in the brine, leading to a higher concentration of CO_2_ and co-crystallization of NH_4_HCO_3_ or NaCl. For this purpose, a small additional absorber (*MAB*) was planned and constructed to increase the ammonization of brine. The diameter of apparatus D = 1000 mm; height of apparatus H = 8000 mm. In the lower segment, there was a brine tank with a volume of V = 0.78 m^3^, with a pre-absorption zone placed on a bed of PALL rings with a contact surface of 219 m^2^. In the middle segment, flow was designed through two layers of cellular filling. The filling was assumed to absorb approximately 65% of the ammonia contained in the gas after passing through the Pall rings. The characteristics of the filling in the middle segment are as follows: process cellular filling with 2 × 3 layers, 316 L, φ600. In the upper segment of the column, a demister was designed, which is a separation cellular filling with three layers, 316 L, φ600.

The additional saturation of brine with ammonia was carried out in such a way that the main stream of brine was dosed to the upper part of the *MAB*. In the counter-current, a gas containing a high concentration of ammonia was fed (over 60% *v*/*v*). The outgoing *AAB* stream was divided into two streams: the first was to be dosed to the carbonization column, and the second returned to the absorption process (return stream). Total alkalinity, the most important *AAB* parameter, depends on two factors: the first is return stream volume and the second is return stream temperature. The values obtained for total alkalinity and the relationship between return stream volume and return stream temperature are presented in Figure 1 and Figure 2.

An increase in return stream volume and a decrease in its temperature result in a higher concentration of ammonia being present in the *AAB* solution. The highest recorded concentration was 159.8 mmol∙20 cm^−3^. In this case, no solid phase precipitation from *AAB* was observed. The tests were carried out on an industrial installation at a technical scale in Ciech S.A. Factory in Minorca.

### 1.2. Carbonization Process

The carbonization of ammonia brine was carried out in a tower apparatus called a carbonization column. It is there that the ammonia brine is saturated in countercurrent with carbon dioxide introduced from the bottom of the column. The chemical transformations that occur in the carbonization column can be summarized in the form of Equation (1), in which the equilibrium is shifted significantly to the left.
(1)$$\mathrm{NaCl}+{\mathrm{NH}}_{4}{\mathrm{HCO}}_{3}\;⇄\;{\mathrm{NaHCO}}_{3}+{\mathrm{NH}}_{4}\mathrm{Cl}$$

This process may seem quite simple, but when considering its individual aspects and complexity, one may come to the conclusion that it is a very complex process, requiring knowledge of issues related to absorption, solubility and crystallization. All the discussed transformations in the reactor take place in three states of aggregation.

Considering the kinetics of the absorption process, one should pay attention to the formation of carbamate ions, considering the equilibrium in the NH_3_-CO_2_-H_2_O system. The analysis of such a system results in the conclusion that, at an elevated temperature (323–333 K), carbamate ions predominate in the studied system, especially when the solution concentration of NH_3_ is higher than that of CO_2_ [11,12]. The reactions of carbamate ion formation can be shown as follows, in Equations (2) and (3).
(2)$${\mathrm{CO}}_{2}+2{\mathrm{NH}}_{3}\;⇄\;{{\mathrm{NH}}_{4}}^{+}+{\mathrm{NH}}_{2}{\mathrm{COO}}^{-}$$


CO_2_ + 2NH_3_ ⇄ NH_2_COO^−^ + H^+^(3)


Therefore, the role that carbamate ions play can be represented as follows, in Equations (4)–(6):(4)$${\mathrm{NH}}_{2}{\mathrm{COONH}}_{4}\;⇄\;2{\mathrm{NH}}_{3}+{\mathrm{CO}}_{2}\;$$
(5)$${\mathrm{NH}}_{3}+H^{+}⇄\;{{\mathrm{NH}}_{4}}^{+}$$
(6)$${\mathrm{CO}}_{2}+{\mathrm{OH}}^{-}\;⇄{\;{{\mathrm{HCO}}_{3}}^{-}}^{\;}$$

As a result, it can be concluded that carbamate ions act as a CO_2_ carrier from the gas phase into solution, and thus condition the formation of HCO_3_^−^ ions. At the time of NaHCO_3_ precipitation, it can also be suggested that the carrier of CO_2_ into the solution meets the NH_3_∙CO_2_ complex, which then decomposes, providing carbamate ion or sodium bicarbonate.

The next step in the process is to obtain the appropriate supersaturation and crystallization of NaHCO_3_. The degree of supersaturation plays a critical role in the crystallization process and depends on several factors, which are analyzed in detail in [13].

To modify the conduction of the carbonization process by increasing the amount of ammonia in the reaction solution, it was necessary to analyze the degree of supersaturation at the start of the crystallization process. The *AAB* stream should be brought to a point in the carbonization column where the ratio of NH_3_ to CO_2_ is unfavorable. Many column-operating parameters had to be analyzed and a number of equilibrium and non-equilibrium analytical studies of the carbonization process had to be performed; this required a large number of calculations. This determination will allow for the skillful dosing of additional ammonia to the carbonization column when there is an ammonia deficiency relative to the carbon dioxide in the carbonization process. To select the reactor-dosing location and the amount of additional ammonia that is to be dosed, it is necessary to determine the non-equilibrium and equilibrium conditions in different parts of the carbonization column and determine the change in ion concentration along the carbonization column. To correctly perform calculations of the non-equilibrium state, it is necessary to use a specific mathematical apparatus and methodology for the mathematical calculation of process parameters, which is all presented in the mathematical relationships section. A description of the tests is presented in the section describing the carbonization column operation tests.

### 1.3. Mathematical Relationships

To calculate the non-equilibrium composition of the carbonization suspension, the generally known relationships, presented in [14], were used. The composition of the non-equilibrium solution is shown through its relationship with the composition of the equilibrium solution; therefore, one should be able to calculate the equilibrium composition. Designations for the non-equilibrium solution will be given with an apostrophe. Calculations concerning the process of *AAB* dosing will be given with the subscript 1. The calculation method of interest to us consists of the ability to relate the composition of the non-equilibrium solution, i.e., parameters: *a*′, *b*′, *c*′, *d*′, *e*′, *f*′, *g*′, *m*′, *n*′, *o*′, *u*′, *w*′, *z*′, with the composition of the corresponding equilibrium solution with parameters *a*, *b*, *c*, *d*, *e*, *f*, *g*, *m*, *n*, *o*, *u*, *w* and *z*.

There are a number of interrelationships between the parameters described above, which allow us to form the following equations:*d*′ = *e*′ + *f*′ + 2*g*′(7)
*m*′ = *n*′ − *d*′ + *g*′(8)
(9)$$K_{2}=\frac{f^{\prime }}{g^{\prime }\cdot m^{\prime }}\;$$
(10)$$R=\frac{2b^{\prime }+d^{\prime }}{a^{\prime }}=\frac{2b+d}{a}{\;}^{\;}$$
(11)$$\frac{a}{a^{\prime }}=\frac{c}{c^{\prime }}{\;}^{\;}$$
*c* = *c*′ [1 + 0.0012(*b* − *b*′)](12)
*a*′ = *n*′ + *b*′(13)
(14)$$z^{\prime }=\frac{b^{\prime }}{c^{\prime }}\;$$

Equation (7) expresses the balance of carbon dioxide content, (8) follows from the definition of alkalinity, and (13) from bound ammonia. The volume of the liquid phase due to the crystallization of NaHCO_3_ decreases by approximately 0.12%, with a precipitation of 0.05 mole of NaHCO_3_·dm^−3^. At the same time, the concentration of components remaining in solution increases. These relations are described by Equations (11) and (12).

Considering the mechanism of the process carried out in the carbonization column, carbamate atoms are formed at a temperature over 343 K, the formation reaction of which can be written in the form:CO_2_ + 2NH_3_ ⇄ NH_4_^+^ + NH_2_COO^−^

Then, under process conditions, the carbamate ion undergoes hydrolysis:NH_2_COO^−^ + H_2_O ⇄ HCO_3_^−^ + NH_3_

This reaction is a limiter of the process and the concentration of carbamate ions deviates from the equilibrium state, marked as “e”. At equilibrium state [e] ≈ 0.

During the transition of the system from the non-equilibrium to the equilibrium state, the degree of carbonization of the system, i.e., the ratio of the total carbon dioxide in solution and in the precipitate to the total ammonia concentration, does not change (Equation (10)).

Equation (9) expresses the equilibrium constant of the reaction:(15)$${\mathrm{HCO}}_{3}^{-}+{\mathrm{OH}}^{-}⇄\;{\mathrm{CO}}_{3}^{2-}+H_{2}O$$

The authors of the cited works assume that this reaction is very fast so that, even in non-equilibrium solution, it reaches an equilibrium state. The numerical value of the constant K_2_ can be calculated from the following equation:(16)$${\mathrm{logK}}_{2}=\frac{-1717}{T}+6.96^{\;}$$

If the composition of the equilibrium solution is known, then, in Equations (10)–(14), there are six unknown values (*a*′, *b*′, *c*′, *d*′, *n*′, *z*′). To determine these, it is necessary to know only one of them. In Equations (7)–(9), after calculating the values of *a*′, *b*′, *c*′, *d*′, *n*′, *z*′, unknown values remain (*e*′, *f*′, *g*′, *m*′); to determine these, only one of them needs to be known.

The following formula is used to calculate the equilibrium pressure of ammonia over an equilibrium solution:(17)$$u^{\prime }=p_{\left(NH_{3}\right)}^{\prime }=\frac{M\cdot e^{\prime }}{g^{\prime }}\;$$
where:(18)$$\mathrm{logM}=\frac{-2064}{T}+7.38^{\;}$$

In the first stage of carbonization, the same M-factor can be used as for equilibrium solutions, i.e., where the ammonia pressures are high, and the non-equilibrium and equilibrium solutions do not much differ from each other. On the other hand, in the second stage of carbonization, the establishment of an equilibrium due to the formation and hydrolysis reactions of carbamate ions should be taken into account [15]. The following formulas can be used to calculate non-equilibrium carbon dioxide pressures:(19)$$w^{\prime }=p_{\left(CO_{2}\right)}^{\prime }=N_{1}\cdot e^{\prime }\cdot {\left(m^{\prime }\right)}^{-2}{\;}^{\;}$$
where:(20)$${\mathrm{logN}}_{1}=\frac{-3390}{T}+13.22^{\;}$$
or:(21)$$w^{\prime }=p_{\left(CO_{2}\right)}^{\prime }=N_{m}\cdot {\left(g^{\prime }\right)}^{2}\cdot \;\frac{a^{\prime }}{d^{\prime }e^{\prime }}\;$$
where:(22)$${\mathrm{logM}}_{m}=\frac{-2400}{T}+8.94^{\;}$$

The crystallization rate of NaHCO_3_ in the process of ammonia brine carbonization depends on the supersaturation of crystallization at a constant temperature, which can be expressed by the following equation:(23)$$W_{k}=K_{k}\cdot (b-b^{\prime })\;\;\left[\mathrm{mmol}\cdot \min^{-1}\right]{\;}^{\;}$$
where:(24)$${\mathrm{logK}}_{k}=\frac{-724}{T}+1.504^{\;}$$

The basis of these calculations is the solution of two independent systems of Equations (7), (9), (10) and (14). This requires knowledge of the concentration of carbamate ions or the sum of carbonate ions. To solve Equations (10)–(14), it is sufficient to know the value of one of the parameters present in the system. It seems that the value of parameter *d*′, i.e., the non-equilibrium concentration of total carbon dioxide, would be optimal. The values of parameters *a*′ and *c*′ do not differ much from the values of a and c. This is very convenient, due to the ease of experimental determination for use in calculations of the value of *n*′, the non-equilibrium alkalinity of the solution; however, this is only possible for the second stage of the carbonization process. The knowledge of the values of parameters *b*′ and *z*′ does not bring anything new, since *b*′ *= a*′ *− n*′ and *z*′ *= b*′/*c*′.

All the conducted studies were performed in accordance with Equations (7)–(24). An interesting feature that influences the efficiency of the soda process is the solution supersaturation. In the soda process, it is important that the supersaturation reaches its maximum value. Supersaturation is defined as:∆*b* = *b* − *b*′

The concentration will be determined in both the equilibrium state and the non-equilibrium state.

The supersaturation value is proportional to the value of carbamate ions produced in the solution, and directly depends on the *m*/*d* and *m*′/*d*′ ratio. The dosing point in the carbonization column H is proportional to the relationship:H ≈ e′*m*/*d*

For the practical use of the analytical model, it is necessary to know the values of the equilibrium constants of the reactions that occur in the NaCl-NH_3_-CO_2_-H_2_O system. In order to determine these values, the CO_2(aq)_ data and the concentration of *g*′, *f*′, *o*′, or *m*′ are necessary. These values are not known; hence, the use of the mathematical apparatus allows for an estimation of the values of the equilibrium constants in the tested system. The analysis and the mathematical apparatus will be the subject of a separate publication.

In sum, it may be stated that, in order to calculate the composition of a non-equilibrium solution, at any temperature and any degree of carbonization, one should have the experimentally determined values of the concentration of carbamate ions or the sum of carbonate ions and total carbon dioxide.

### 1.4. Tests of Carbonization Column Operation

Investigations into the course of the carbonization process, under the actual conditions, at the technical scale, were undertaken to determine the possibility of improving the parameters of the carbonization column to increase the efficiency of the process and improve the quality of the obtained precipitate. The research was conducted on column No. 10 at the Soda Ash Factory in Inowrocław, Poland. The column scheme is shown in Figure 3.

The investigations were carried out on the column, as shown in Figure 3. Column height (h_k_) was 28.7 m, and the height of the upper part (h_g_) was 15.8 m. In Figure 1, the following is indicated:*L_s_*: brine flow rate;*G_a_*: exhaust gas flow rate;*G*_1_: upper gas flow rate;*G*_2_: bottom gas flow rate;*W*: cooling water flow rate;*t_s_*: temperature of the brine fed;$tg_{1}$: upper gas temperature;$tg_{2}$: bottom gas temperature;$tw_{1}$: cooling water outlet temperature;$tw_{0}$: cooling water inlet temperature.

The study methodology consists of a minimum two-hour process to stabilize the carbonization column under appropriately selected conditions. After this time, the suspension samples were taken at specific time intervals. For this purpose, the column was equipped with sampling taps at the following locations: barrel 9, barrel 7, barrel 3, box 7, box 4, box 2, and sodium bicarbonate suspension outlet. A total of seven sampling points were selected along the carbonation column zones. Sampling vessels were thermostated, and analyses were started to minimize the effect of time on the sample processes after collection. The sample was protected by dilution and analyzed for *n*, *a*, *b*, *c* and *d* in the process laboratory.

Samples were directly collected from the in-process carbonization column as a post-reaction suspension of bicarbonate at room temperature. Immediately after collection, the suspension was filtered under reduced pressure on a Büchner funnel. After the separation of solid and liquid, the precipitate was washed with anhydrous methanol and dried. After drying, the obtained sodium bicarbonate sample was secured in an airtight container.

## 2. Materials and Methods

The identification of the suspension included the determination of direct alkalinity, total alkalinity (*n*), the amount of CO_2_ (*d*), content of chloride ions (Cl^−^) (*c*), and ammonium chloride (NH_4_Cl) (*b*).

The method involved distilling off the ammonia and absorbing it in a standard solution of sulfuric (VI) acid. Excess sulfuric (VI) acid was titrated with a standard solution of sodium hydroxide in the presence of methyl orange. Ammonia contained in liquids in the form of carbonate and bicarbonate salts passed into the gas phase due to the thermal decomposition of these compounds; this is called free ammonia.

Bound ammonia, in the form of NH_4_Cl, decomposes with a strong alkali such as sodium hydroxide.

### 2.1. Direct Alkalinity (n)

To determine alkalinity, the sample was titrated with 0.5 M sulfuric (VI) acid in the presence of methyl orange. The titration was carried out until the color of the solution changed from yellow to orange–yellow. The alkalinity value along the carbonization column is shown in Table 1.

### 2.2. Amount of CO_2_ (d)

The determination was performed on a Scheibler apparatus. The carbonate and bicarbonate ions contained in the sample were decomposed using hydrochloric acid with the release of free CO_2_, which displaced the liquid from the gas burette tube so that the amount of gas released can be determined. The amount of CO_2_ along the carbonization column is shown in Table 1.

### 2.3. Content of Cl^−^ (c)

The determination of chloride ions was performed using a compact potentiometric titrator Metrohm Ti-Touch 916. To perform the measurement, the sample was neutralized with sulfuric (VI) acid, and then diluted and acidified with the same acid. The sample was then titrated with 0.03333 M silver nitrate solution in the presence of a silver electrode, using the appropriate instrument software. The chloride anions content along the carbonization column is shown in Table 1.

### 2.4. Content of NH_4_Cl (b)

A potentiometric titrator was used to determine the amount of ammonium chloride in the sample. To carry out this measurement, the sample was boiled in lye. For this, the sample was added to a specific amount of sodium hydroxide and then allowed to stand on a hot plate to boil off the ammonia. The remaining excess alkali was titrated with 1 M sulfuric (VI) acid in the presence of a glass electrode using the appropriate program for the instrument. The ammonium chloride content along the carbonization column is shown in Table 1.

### 2.5. Content of NH_2_COONH_4_ (e)

The presence of carbamate ions was confirmed by ^13^C NMR (Bruker Scientific LLC, Billerica, MA, USA) technique; the content of carbamate ions was analyzed using IR spectrometry (Bruker Scientific LLC, Billerica, MA, USA) at a wavelength 687 cm^−1^. The ammonium carbamate content along the carbonization column is shown in Table 2 and Figure 4.

### 2.6. Solid Phase Identification

To identify the prepared solid phase, TG-DSC (analysis was performed using a Jupiter STA 449 F5 thermoanalyzer (Netzsch, Selb, Germany) with gaseous decomposition products identified in a Vertex 70v infrared spectrometer from Brüker Optik coupled to the thermoanalyzer. To perform this analysis, a stable sample was exposed to temperature. During this time, the thermoanalyzer time and temperature-dependent changes in the mass of the sample were recorded. The decomposition products were transported by capillary to the IR spectrometer, where they were analyzed in the infrared absorption spectra.

## 3. Results

Example results for the column operation are shown in Table 1.

On this basis, it is possible to visualize the sodium yield of the process as a function of the place at which the additionally ammonized brine is dosed (Figure 5). To do this, it is necessary to define the term soda yield in the process.

As can be seen from reaction (1.1), the amount of NaHCO_3_ formed in the soda process is equivalent to the amount of the resulting NH_4_Cl. However, part of the resultant NaHCO_3_, determined by the solubility product under given conditions, is found in solution. The amount of precipitated NaHCO_3_ in the form of precipitate will, therefore, be (25):NaHCO_3 (s)_ = NH_4_Cl − NaHCO_3 (aq)_(25)

It is known that:alkalinity = NaHCO_3 (aq)_ + NH_4_HCO_3_(26)
and
NH_3 (total)_ = NH_4_Cl + NH_4_HCO_3_
(27)
therefore:NaHCO_3 (s)_ = NH_3 (total)_ − alkalinity(28)

To calculate the efficiency of the carbonization process relative to Na^+^ ions, it is necessary to determine the ratio of Na^+^ ions bound as NaHCO_3_ present in the precipitation to the total amount of sodium ions introduced into the process as NaCl. Therefore, the efficiency of the carbonization process was calculated based on the formula:(29)$$W_{Na}=\frac{{\mathrm{NH}}_{3\;\left(\mathrm{total}\right)}-\mathrm{alkalinity}}{{\mathrm{Cl}}^{-}}$$

The equilibrium results of the carbonization process are presented in Table 1. As can be seen from the results and calculations, the additional *AAB* stream should be directed to the area of the reactor where the concentration of the intermediate product, i.e., ammonium carbamate, is the lowest or quickly reaches equilibrium. This situation occurs at the top of the carbonization column. Here, the synthesis of carbamate ions occurs at a high rate, and consequently, the driving force of the crystallization process is slightly lower. The concentration of ammonium carbamate was investigated in both equilibrium and non-equilibrium states; the results are summarized in Table 2 and presented in Figure 4. From the conducted experiments, it is evident that the optimum *AAB* dosing location is barrel 10, which is confirmed by the obtained sodium yield values of the process. It is noteworthy that the concentration of ammonium carbamate at equilibrium increased at the top of the column. Higher values of NH_2_COONH_4_ concentration are maintained up to the fourth cooling box. Below this, the ammonium carbamate concentration reaches the values of the tests conducted without dosing the additional *AAB* stream.

Interpreting the test results for the plant’s technological process and the modifications, it was also found that increasing the concentration of ammonia is necessary when the temperature at the outlet of the carbonization column is higher than that in a typical technological process. It follows that the decrease in process efficiency with increasing column outlet temperature can be prevented if the amount of ammonia in the circuit is increased. This is the second positive aspect of dosing extra amounts of additionally ammonized brine. It also minimizes energy consumption, e.g., for column cooling during summer operation. Table 3 provides a summary, and Figure 6, Figure 7, Figure 8, Figure 9, Figure 10 and Figure 11 present process parameter values for carbonization column measured at the column outlet during *AAB* dosing and without *AAB* dosing. As shown in Figure 6, in all the presented cases, the soda process yield was higher during *AAB* dosing, as was the direct alkalinity (Figure 8), which is consistent with soda process theory. The mean value of soda yield without dosing was 75.676% with a standard deviation of 0.749%; with dosing 77.482% with a standard deviation of 0.621% (Figure 7). The mean value of direct alkalinity without dosing was 22.57 mmol·20 cm^−3^ with standard deviation 0.82 mmol·20 cm^−3^; with dosing this was 27.80 mmol·20 cm^−3^ with standard deviation 2.09 mmol·20 cm^−3^ (Figure 9).

The concentration of chloride ions during the carbonization process increases along the carbonization column, which is due to the desorption of water from the system and the consequent concentration of the carbonization solution. When dosing an extra stream of additionally ammonized brine, the solution volume is increased by increasing the ammonia concentration. The consequence is a decrease in the solubility of NaCl, and to keep this constant, the degree of CO_2_ absorption must be increased. Under the operating conditions of the plant, it is not possible to saturate the solution with sodium chloride. Therefore, according to Equation (29), the process efficiency increases. Selected values of chloride ion concentration in carbonization solution are summarized in Table 3 and presented in Figure 10. Results 6 and 10 differ from the expected values; they are actual results. The mean value of chloride ion concentration in the solution without *AAB* dosing is 98.10 mmol·20 cm^−3^ with the standard deviation of 0.31 mmol·20 cm^−3^. The mean value of the chloride ion concentration in the solution with *AAB* dosing is 96.71 mmol·20 cm^−3^, with a standard deviation of 1.12 mmol·20 cm^−3^. The difference in mean values is 1.39 mmol·20 cm^−3^, which coincides with the expected value (Figure 11). It should be noted that the increase in efficiency is a result of an increase in total ammonia concentration and decrease in chloride ions concentration in the process. Ammonium chloride content during *AAB* dosing, considering mean values, also increases (Figure 12). The mean value of ammonium chloride concentration in the solution without *AAB* dosing is 74.24 mmol·20 cm^−3^, with a standard deviation of 0.81 mmol·20 cm^−3^. The mean value of ammonium chloride concentration in the solution with *AAB* dosing is 74.93 mmol·20 cm^−3^, with a standard deviation of 0.96 mmol·20 cm^−3^. The difference in mean values is 0.69 mmol·20 cm^−3^, which coincides with the expected value (Figure 13). When dosing *AAB*, the total ammonia content significantly increases (Figure 14), which is obvious because there is an additional amount of NH_3_ in the system. All recorded values of total ammonia concentration are higher than the values of this parameter without *AAB* dosing to B10. The mean value of the total ammonia concentration in solution without *AAB* dosing is 96.81 mmol·20 cm^−3^, with a standard deviation of 0.94 mmol·20 cm^−3^. The mean value of the total ammonia in solution with *AAB* dosing is 102.7 mmol·20 cm^−3^, with a standard deviation of 1.6 mmol·20 cm^−3^. The difference in the mean values is 5.921 mmol·20 cm^−3^ (Figure 15), which is the result of the additional ammonia entering the reactor and the amount desorbed under different hydrodynamic conditions. During *AAB* dosing, the temperature of the lower cooling section increases. This is very important, considering the seasonality of the process. In summer, when the cooling water temperature is high, it is not possible to sufficiently cool the carbonization column. By dosing appropriate amounts of *AAB* in a designated area of the reactor, it is possible to maintain adequate efficiency of the process under worse cooling conditions. Under winter conditions, the temperature at the bottom of the carbonization column is about 293–303 K. In summer conditions, this often reaches the temperature of 313 K, which decreases the efficiency of the carbonization process.

The increased total ammonia content has another important consequence. While increasing the ammonia concentration and wishing to maintain the process parameters, especially without changing the solubility of sodium chloride, the concentration of carbon dioxide in solution must be increased. Under *AAB* dosing to the carbonization column, the solubility of CO_2_ in solution increases, as is primarily seen by the increase in the degree of carbonization of the solution. By absorbing larger amounts of CO_2_ from the gas fed to the reactor, we reduce the amount of loss of CO_2_ in the soda process, which is one of the main environmental effects of the changes made when dosing the additional *AAB* stream. The total carbon dioxide content in solution (d), determined during the technical scale experiment, is summarized in Table 3 and shown in Figure 16. The mean concentration of total carbon dioxide in solution without *AAB* dosing is 91.13 mmol·20 cm^−3^, with a standard deviation of 5.726 mmol·20 cm^−3^. The mean concentration value of total carbon dioxide in solution with *AAB* dosing is 94.91 mmol·20 cm^−3^, with a standard deviation of 7.61 mmol·20 cm^−3^. The difference in mean values is 3.78 mmol·20 cm^−3^. As shown in Figure 17, there are no significant differences between the mean values of total CO_2_ in solution with and without *AAB* dosing. One can see a slight advantage in favor of dosing. As the solution was diluted and was more alkaline, it is obvious that more CO_2_ dissolves during *AAB* dosing. To accurately quantify additionally dissolved CO_2_, one should consider the dilution factor of the solution, which can be calculated by determining the concentration of chloride ions.

Conducting research work at a full technical scale, the effects of ecological activities, such as dosing an additional stream of *AAB* to the carbonization column, were determined with high accuracy. The most important ones are an increase in productivity by increasing the carbonization degree and consequent increases in the binding of CO_2_, reduction in ammonia, and carbon dioxide losses in the exhaust gases and energy savings due to the lower cooling of the carbonization column. Changes in CO_2_ losses over the course of the experiment are shown in Figure 18. The average CO_2_ concentration in the exhaust gas during the 58-h test was 7.34% *v*/*v* for *AAB* dosing and 12.21% *v*/*v* for the experiment without *AAB* dosing. The difference is 4.87% *v*/*v*, which represents a significant reduction in emissions. The result is logical and is due to the reactivity and equilibrium realization of the carbonization process.

For CO_2_ and NH_3_ concentration control, an Ultramat 23 gas analyzer by SIEMENS K3-374 was installed in the industrial installation at the outlet of the gases from the column. The automation of sampling, drying, and the use of IR analysis made it possible to measure the concentration of CO_2_ and NH_3_ in the exhaust gases.

The analyzer is calibrated in two points: with pure nitrogen and a mixture of 20% CO_2_ in nitrogen.

A positive experimental result was also found when analyzing the productivity of the carbonization column. During the test without *AAB* dosing (normal operation of the reactor), the mean productivity was determined, which was 11.91 Mg NaHCO_3_·h^−1^, 285.80 Mg·d^−1^, 690.68 Mg·cycle^−1^. During the test, with the dosing of additionally ammonized brine, the analyzed productivity was: 13.27 Mg NaHCO_3_·h^−1^, 318.40 Mg·d^−1^, 769.47 Mg·cycle^−1^ (Figure 19). Several factors contribute to the carbonization column productivity.

The carbonization process efficiency and process speed are the main factors that directly affect productivity. The efficiency of the carbonization process is improved by increasing the carbonization rate and the speed of the process can be increased by better cooling parameters. That is, the carbonization column does not need to be cooled to such a low temperature to maintain adequate soda yield during the process. Detailed results of example production cycles, with and without *AAB* dosing, are included in the appendix of this article.

An important environmental factor and parameter affected by the technique modifications presented in this work is CO_2_ emissions. The structure of carbon dioxide emissions in CIECH Soda Polska for the Soda Ash plant in Inowrocław for the production of soda ash is presented in Figure 20.

Global soda ash production in 2021 was 59 Tg, of which 17 Tg was mine production and 42 Tg was typical Solvay process production. The output of the others is negligible. The soda industry is considered highly disruptive to the environment, and consequently, large effluent gas emissions to the atmosphere make it worth developing and spending time modifying existing technology to reduce the environmental effect of soda production. The assumptions made in the presented work are based on the equilibrium and non-equilibrium state of the AB carbonation process. The equilibrium of the analyzed process is difficult to achieve in process conditions; variability in production, substrate supply and uncontrolled emission are factors with which one often has to struggle in industrial conditions.

The factor that contributes the most to the production of CO_2_ in the soda process is the combustion of coke. The value of the contribution of the balance in CO_2_ to this process is 0.54 Mg CO_2_ per Mg of soda produced. The structure of CO_2_ utilization is dominated by soda production, and the largest contribution is 0.41 Mg CO_2_ per Mg of soda produced. Balancing the process in annual terms, the emissions of CO_2_ to the environment is 0.277 Mg CO_2_ per Mg of soda produced. Considering an annual production of 760 Gg of soda, the annual CO_2_ emissions are 210.5 Gg CO_2_. With the calculated difference in gas emissions from the carbonation column, and assuming a reduction of 4.87% for the entire soda production, the emission reduction for CO_2_ would be 7.93 Gg CO_2_ per year. With the present cost of emissions [16] in Poland at 90 EUR per Mg CO_2_, the reduction in costs is 713.700 EUR per year. It should be noted that CO_2_ emission prices have an upward trend, so any modification to the soda production technique that contributes to both environmental and financial savings has a chance of industrial implementation.

## 4. Conclusions

The authors of this paper developed the theoretical assumptions of the problem of increasing ammonia in the production of sodium bicarbonate and applied them at the technical scale. The additional amount of ammonia in the system is important for the efficiency of the process, which was carried out under certain physical and chemical conditions. In the tested concentration range, this did not affect the composition of the solid phase of the product. It was possible to improve the sodium efficiency of the process by 1.806%, which has a significant impact on the production volume: an increase of 20.8 Gg/year for the entire plant. The main effect of the research is a reduction in CO_2_ emissions from the carbonization process by 4.87%. At the scale of the plant at which the experiment was carried out, this means a reduction of CO_2_ emissions of 7.93 Gg per year, which is a big ecological effect. Dosing an additional *AAB* stream also allowed for water consumption to be reduced for the cooling process of the carbonization column, and thus reduced energy expenditure. The results obtained for the analyzed process show its potential for use for modifications in the whole production plant, and possibly in soda plants all over the world.

## Figures and Tables

**Figure 1 materials-15-04828-f001:**
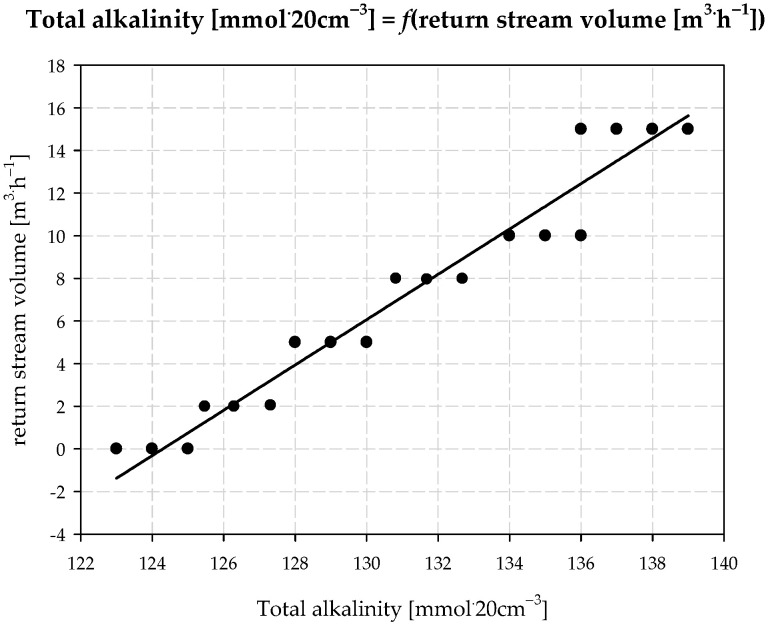
Effect of return stream volume on total alkalinity of *AAB*.

**Figure 2 materials-15-04828-f002:**
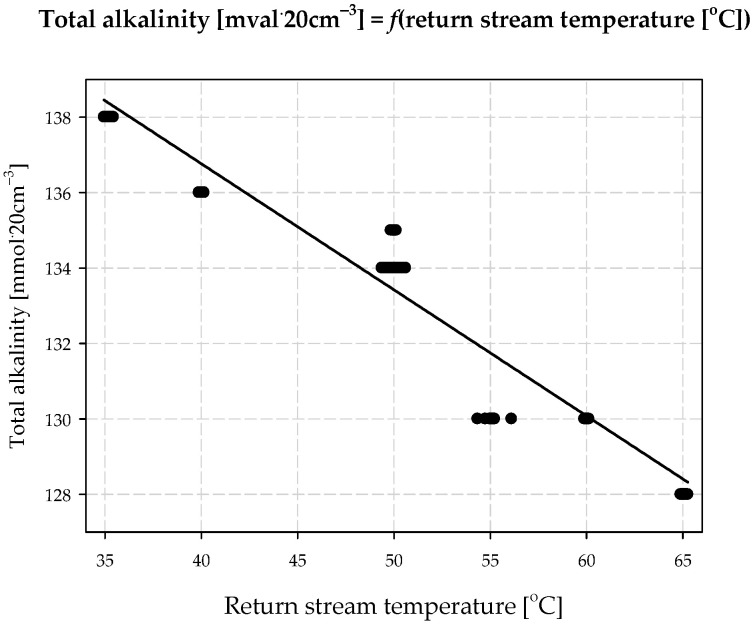
Effect of return stream temperature on total alkalinity of *AAB*.

**Figure 3 materials-15-04828-f003:**
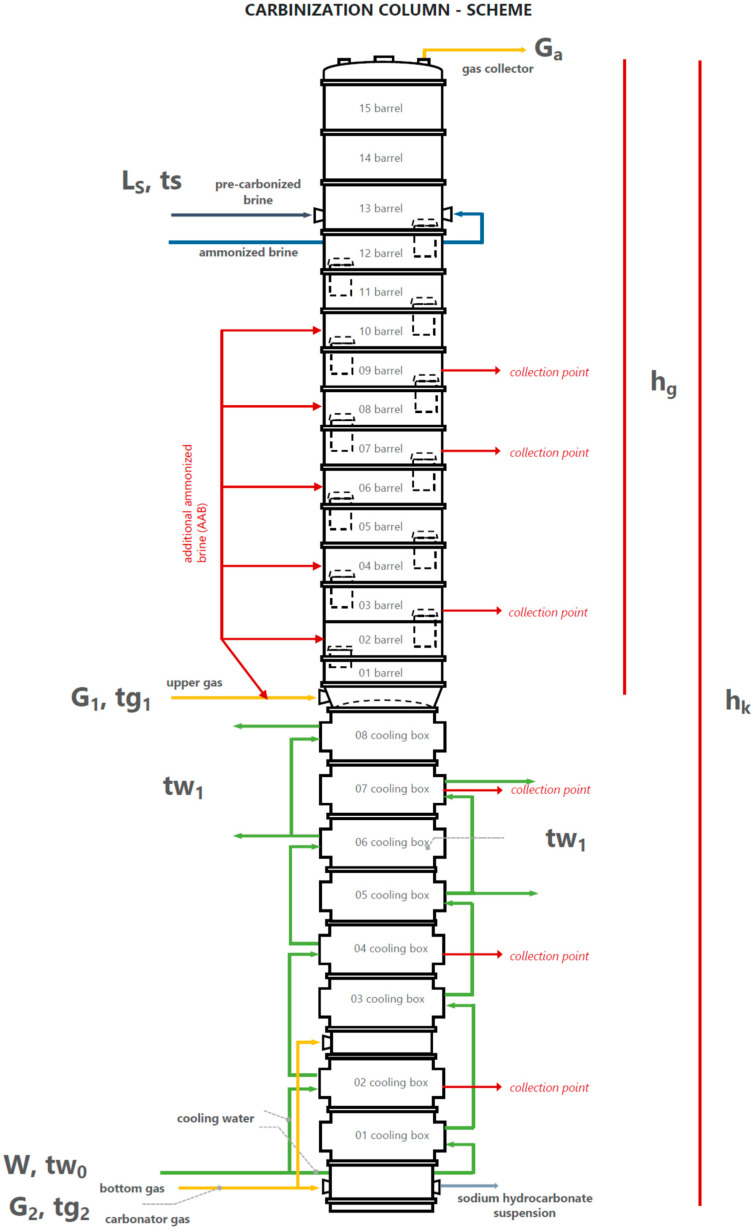
Production/research scheme of the carbonization column.

**Figure 4 materials-15-04828-f004:**
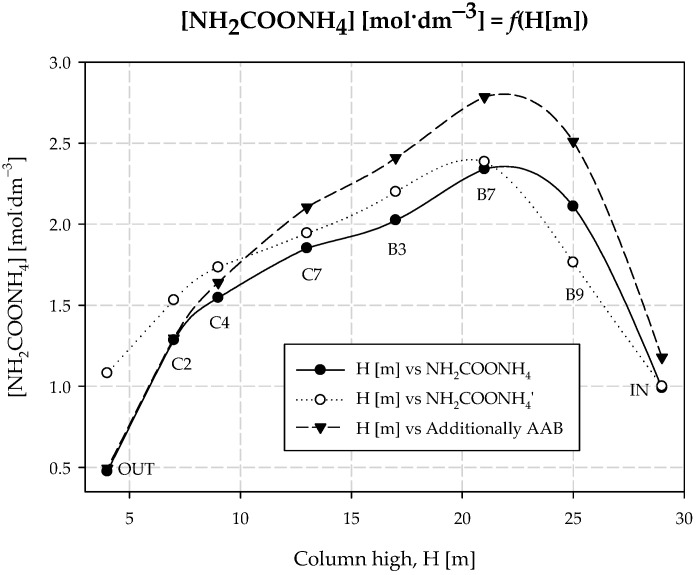
Concentration of ammonium carbamate along the carbonization column.

**Figure 5 materials-15-04828-f005:**
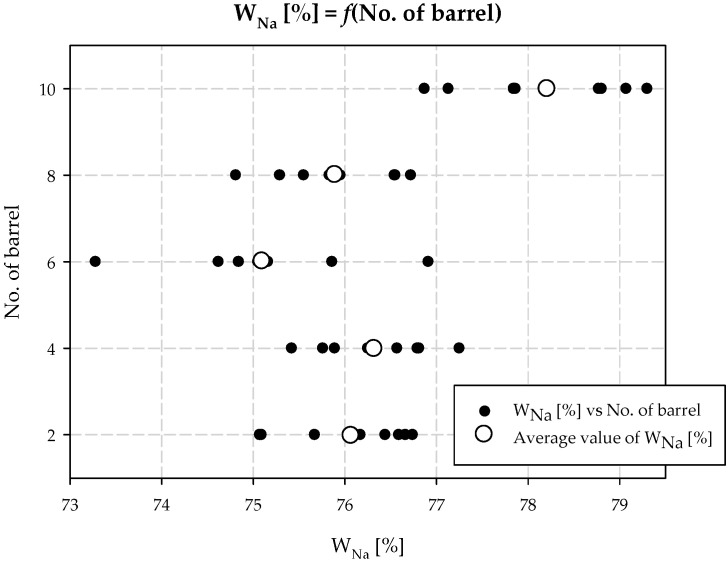
Carbonization yield and *AAB* dosing place (No. of barrel).

**Figure 6 materials-15-04828-f006:**
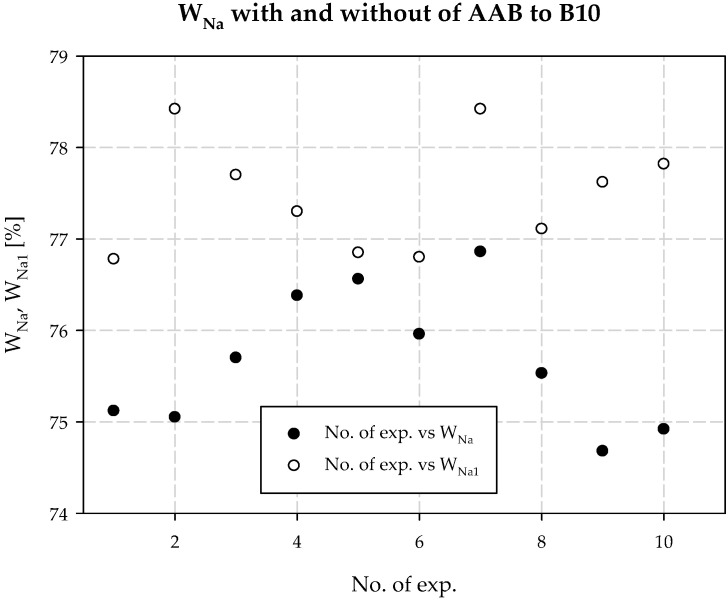
*W_Na_* with and without dosing of *AAB* to B10.

**Figure 7 materials-15-04828-f007:**
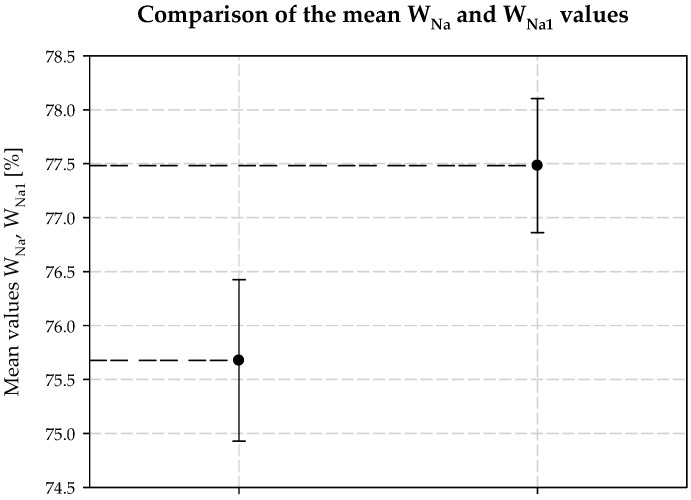
Mean *W_Na_* and *W_Na1_* values with and without dosing *AAB* to B10.

**Figure 8 materials-15-04828-f008:**
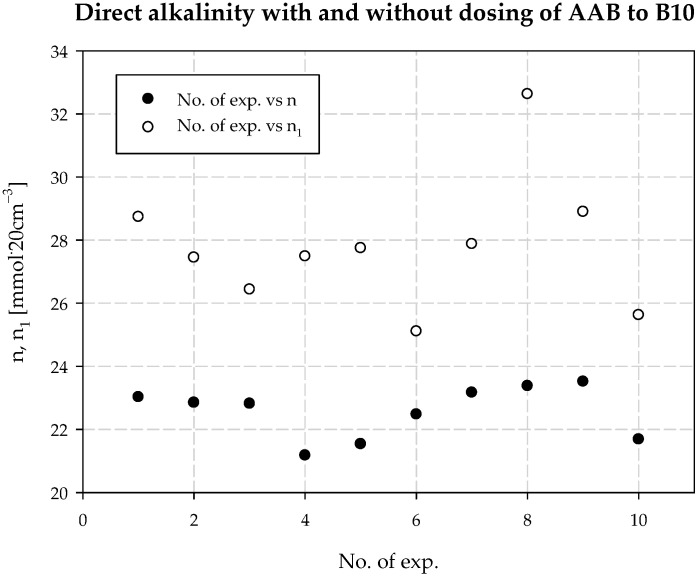
*n* with and without dosing of *AAB* to B10.

**Figure 9 materials-15-04828-f009:**
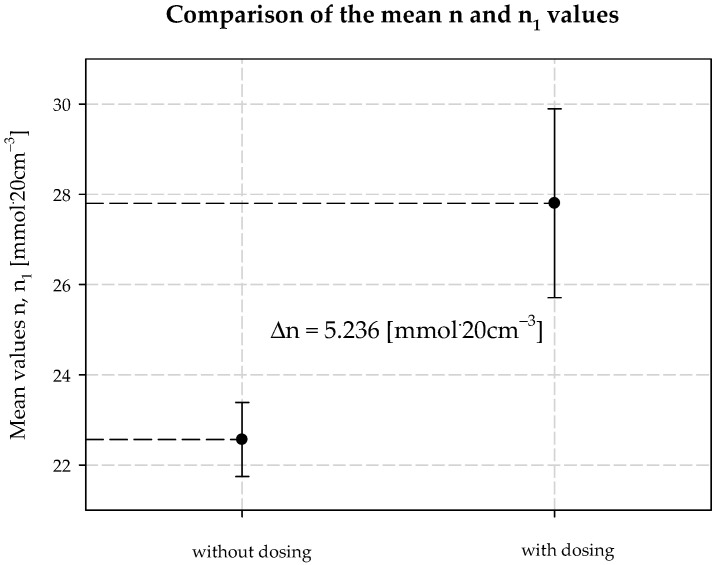
Direct alkalinity values with and without dosing *AAB* to B10.

**Figure 10 materials-15-04828-f010:**
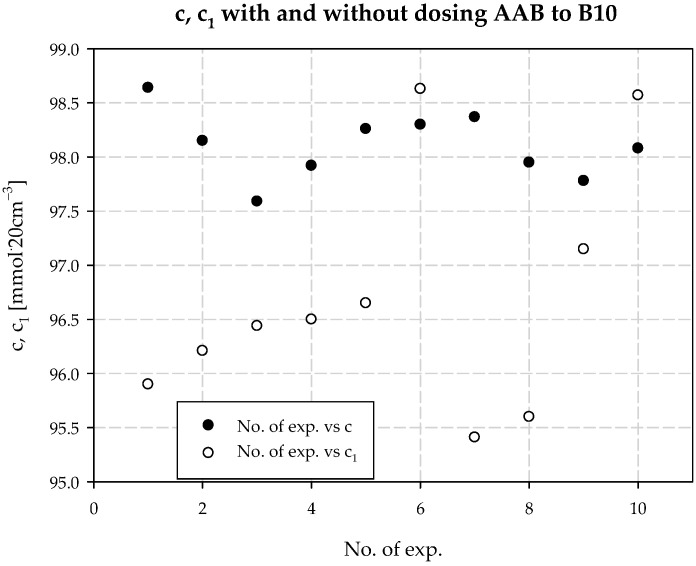
*c* with and without dosing of *AAB* to B10.

**Figure 11 materials-15-04828-f011:**
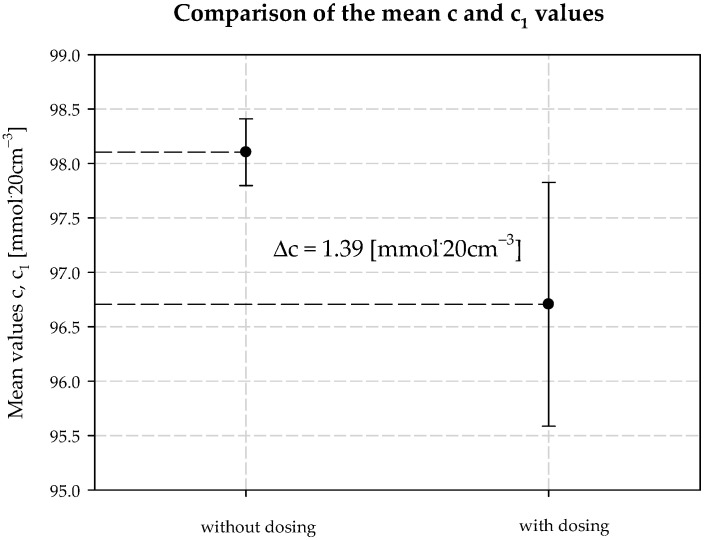
Chlorides concentration values with and without dosing *AAB* to B10.

**Figure 12 materials-15-04828-f012:**
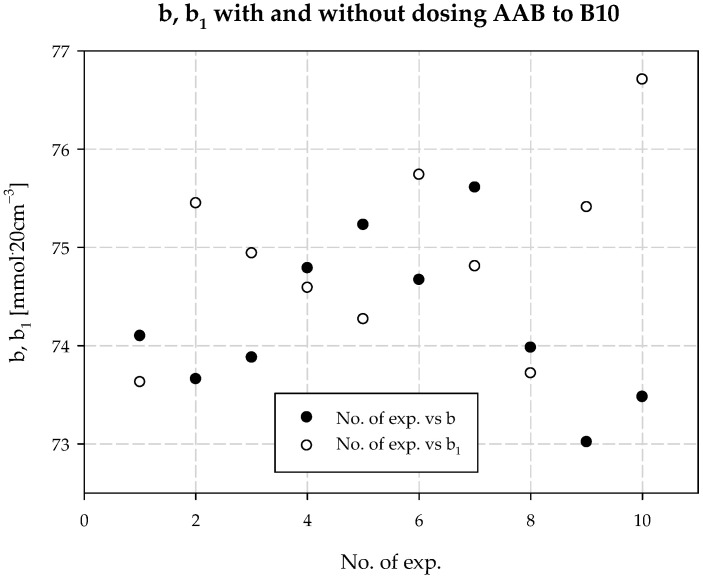
*b* with and without dosing of *AAB* to B10.

**Figure 13 materials-15-04828-f013:**
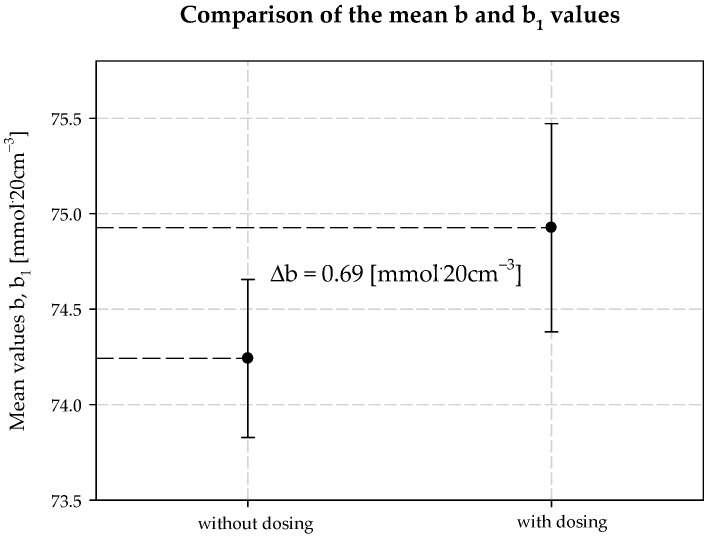
Ammonium chloride concentration values with and without dosing *AAB* to B10.

**Figure 14 materials-15-04828-f014:**
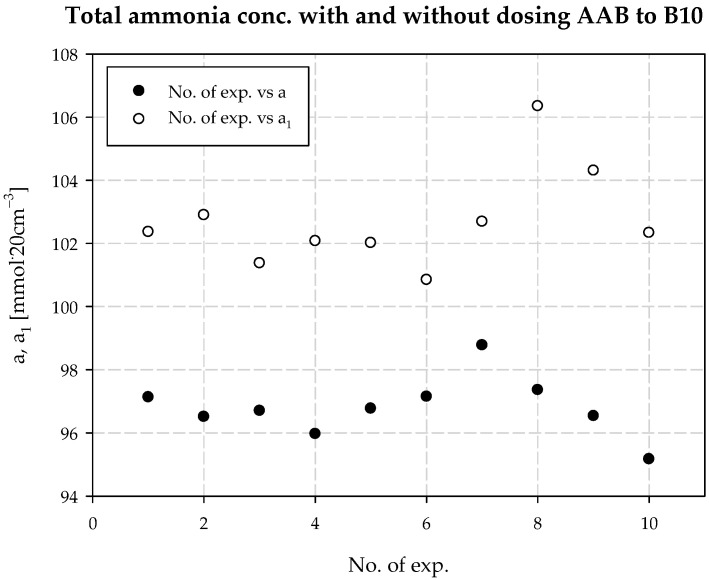
*a* with and without dosing of *AAB* to B10.

**Figure 15 materials-15-04828-f015:**
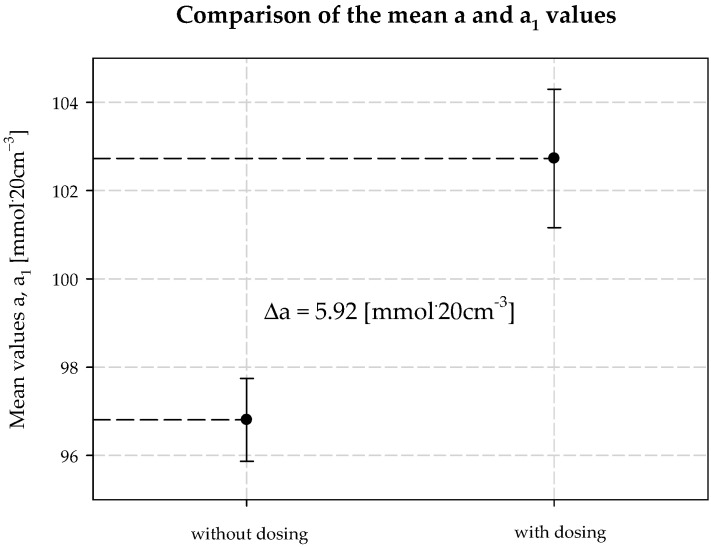
Total ammonium concentration values with and without dosing *AAB* to B10.

**Figure 16 materials-15-04828-f016:**
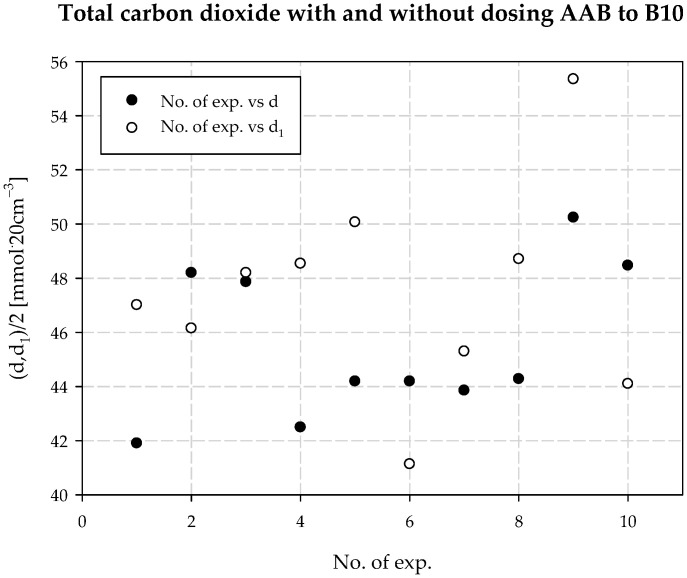
*d* with and without dosing of *AAB* to B10.

**Figure 17 materials-15-04828-f017:**
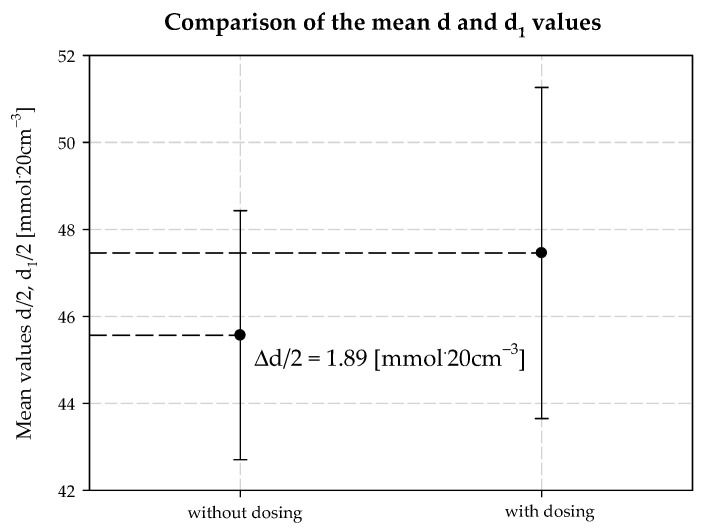
Total carbon dioxide concentration values with and without dosing *AAB* to B10.

**Figure 18 materials-15-04828-f018:**
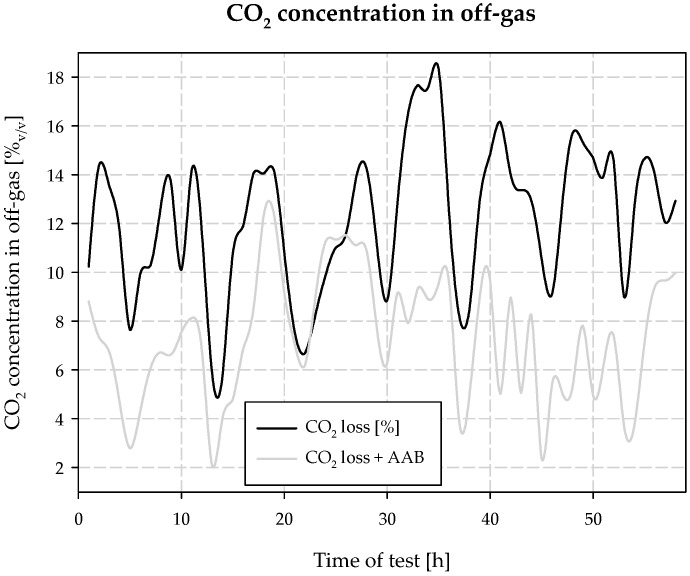
CO_2_ losses during the test of dosing the *AAB* to barrel 8.

**Figure 19 materials-15-04828-f019:**
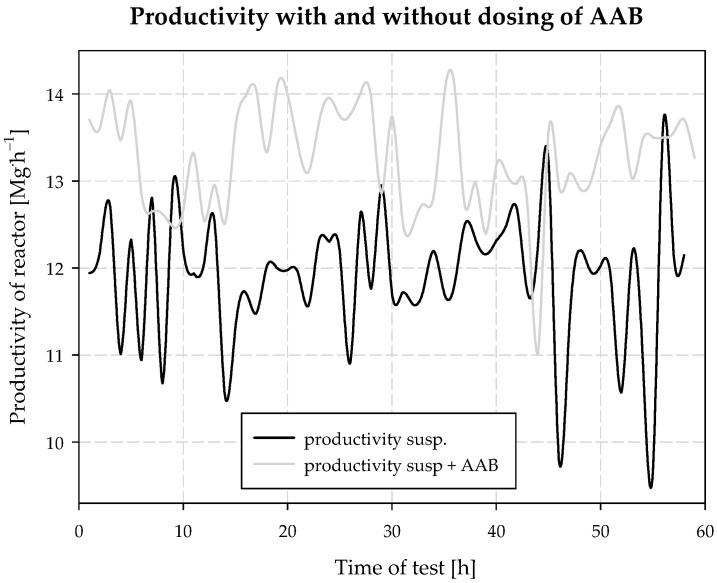
Productivity of reactor with and without dosing of *AAB*.

**Figure 20 materials-15-04828-f020:**
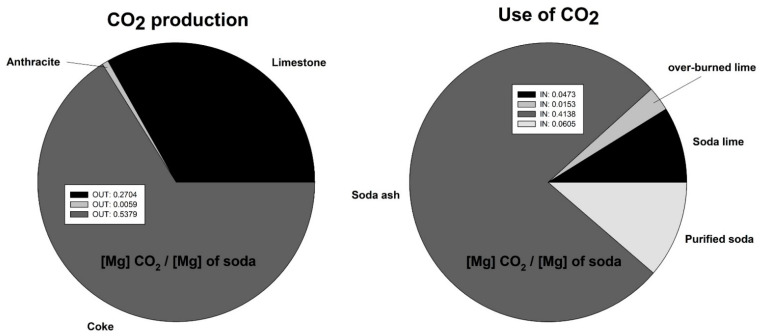
Production and use of CO_2_ in Inowroclaw factory for soda ash process.

**Table 1 materials-15-04828-t001:** Selected results of dosing an additional stream of ammonia to the selected place of the column.

Column High	Date	Hour	*n*	*c*	*b*	*a*	*d*	*W_Na_* [%]
			[mmol 20 cm^−3^]					
10 barrel	18 October 2018	12:02	24.58	94.74	75.13	99.71	43.89	79.30
	18 October 2018	14:25	23.52	95.06	75.17	98.69	39.55	79.07
	18 October 2018	16:30	23.64	95.04	74.89	98.53	39.05	78.80
	18 October 2018	19:48	23.84	95.06	74.88	98.72	40.05	78.77
	19 October 2018	00:14	24.11	95.81	74.58	98.69	40.55	77.84
	19 October 2018	04:07	24.92	91.85	71.51	96.43	40.38	77.86
	3 November 2018	20:55	23.73	95.79	73.88	97.61	43.39	77.13
	4 November 2018	12:04	25.98	95.41	73.34	99.32	44.89	76.87
8 barrel	21 November 2018	19:55	26.26	96.22	73.66	99.92	42.22	76.55
	22 November 2018	00:10	27.06	96.24	73.66	100.72	45.22	76.54
	22 November 2018	04:10	27.62	92.57	70.20	97.82	47.22	75.83
	22 November 2018	08:13	31.50	91.26	68.71	100.21	51.56	75.29
	22 November 2018	12:04	28.76	92.20	70.74	99.50	40.47	76.72
	22 November 2018	16:30	29.46	93.55	69.98	99.44	36.21	74.81
	22 November 2018	20:03	28.49	96.65	73.02	101.51	37.54	75.55
	22 November 2018	21:48	29.76	96.21	73.07	102.83	37.54	75.95
6 barrel	14 June 2018	10:50	27.17	96.48	74.20	101.37	48.88	76.91
	14 June 2018	14:00	25.28	97.70	73.43	98.71	42.66	75.16
	14 June 2018	16:00	25.93	97.91	73.27	99.20	40.62	74.84
	14 June 2018	20:00	27.83	97.05	72.87	100.70	40.87	75.08
	14 June 2018	23:55	27.33	97.13	72.48	99.81	42.41	74.62
	15 June 2018	04:00	26.98	96.97	73.56	100.54	42.49	75.86
	15 June 2018	07:38	31.50	96.14	72.15	103.65	42.07	75.05
	15 June 2018	10:03	33.74	95.94	70.30	104.04	43.34	73.28
4 barrel	21 June 2018	09:30	25.89	96.06	73.25	99.14	40.38	76.25
	21 June 2018	16:00	24.40	96.98	74.49	98.89	41.05	76.81
	22 June 2018	00:00	26.22	97.42	74.60	100.82	40.47	76.57
	22 June 2018	04:00	25.57	97.60	74.95	100.52	38.38	76.79
	22 June 2018	07:51	26.85	95.57	73.83	100.68	44.89	77.25
	22 June 2018	10:10	26.43	97.87	74.15	100.58	46.56	75.76
	22 June 2018	12:35	22.96	96.73	73.41	96.37	37.54	75.89
	22 June 2018	14:15	24.57	97.23	73.33	97.90	38.21	75.42
2 barrel	27 June 2018	16:00	24.03	98.37	73.86	97.89	40.05	75.09
	28 June 2018	07:24	27.46	96.82	74.22	101.68	46.81	76.66
	28 June 2018	08:50	27.75	97.19	74.43	102.18	42.05	76.59
	28 June 2018	11:21	27.77	96.79	74.27	102.04	45.05	76.74
	28 June 2018	16:00	27.77	95.07	72.41	100.18	45.22	76.17
	28 June 2018	20:00	24.90	95.94	72.60	97.50	43.22	75.67
	29 June 2018	00:00	26.22	95.63	71.79	98.01	45.05	75.07
	29 June 2018	04:00	25.66	96.64	73.87	99.53	44.55	76.44

**Table 2 materials-15-04828-t002:** Ammonium carbamate concentration (equilibrium and non-equilibrium) along the carbonization column.

Column High [m]	NH_2_COONH_4_ [mmol·dm^−3^]	NH_2_COONH_4_’ [mmol·dm^−3^]
4	0.9900	1.0000
7	2.1100	1.7650
9	2.3400	2.3850
13	2.0250	2.2000
17	1.8520	1.9450
21	1.5460	1.7350
25	1.2850	1.5320
29	0.4750	1.0820

**Table 3 materials-15-04828-t003:** Selected results with and without dosing *AAB* to carbonization column.

Process	Date	Hour	*n*	*c*	*b*	*a*	*d*	*W_Na_* [%]
			[mmol 20 cm^−3^]					
Without dosing *AAB*	7 June 2018	10:27	23.03	98.64	74.10	97.13	41.90	75.12
	7 June 2018	11:41	22.85	98.15	73.66	96.51	48.20	75.05
	7 June 2018	12:33	22.82	97.59	73.88	96.70	47.86	75.70
	7 June 2018	16:00	21.18	97.92	74.79	95.97	42.49	76.38
	7 June 2018	20:00	21.54	98.26	75.23	96.77	44.19	76.56
	8 June 2018	00:00	22.48	98.30	74.67	97.15	44.19	75.96
	8 June 2018	04:00	23.17	98.37	75.61	98.78	43.85	76.86
	8 June 2018	07:27	23.38	97.95	73.98	97.36	44.28	75.53
	8 June 2018	09:18	23.52	97.78	73.02	96.54	50.24	74.68
	8 June 2018	10:38	21.69	98.08	73.48	95.17	48.47	74.92
Without dosing *AAB*	4 June 2018	10:45	28.74	95.90	73.63	102.37	47.01	76.78
	4 June 2018	12:30	27.45	96.21	75.45	102.90	46.15	78.42
	4 June 2018	16:00	26.44	96.44	74.94	101.38	48.20	77.70
	4 June 2018	20:00	27.49	96.50	74.59	102.08	48.54	77.30
	5 June 2018	00:00	27.75	96.65	74.27	102.02	50.07	76.85
	6 June 2018	04:00	25.11	98.63	75.74	100.85	41.13	76.80
	6 June 2018	07:24	27.88	95.41	74.81	102.69	45.30	78.42
	6 June 2018	09:23	32.63	95.60	73.72	106.35	48.71	77.11
	6 June 2018	11:23	28.90	97.15	75.41	104.31	55.35	77.62
	6 June 2018	13:34	25.63	98.57	76.71	102.34	44.10	77.82

## Data Availability

Data sharing is not applicable to this article.

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
