# Peer review of "Changes in Synthetic Soda Ash Production and Its Consequences for the Environment"

_materials, 2022, doi:10.3390/ma15144828_

Round 1

Reviewer 1 Report

This paper deals with interesting topic on the minimize the environmental burden and minimize the production input of Na2CO3 based on a modification of the soda process. The authors conducted valuable research where was indicated a process that can reduce emissions of CO2 and increased the efficiency of the synthetic soda production, resulting in a measurable financial benefit. The organization of the manuscript is good. I suggest that this paper can be accepted, but I suggest some revisions that should be incorporated on the paper:

1- The introduction needs to be expanded, indicating the art state of the synthetic soda production.

2- Where the data utilized to create the Fig. 1 and 2 were collected? Please insert this information on the paper.

3- Its important to introduce in the conclusion the main contribution of the paper. It was not clear.

Author Response

Dear reviewer,
we tried to answer all questions. We have followed all comments. Detailed file attached.

Best regards
Marcin Cichosz and Authors

Reviewer 2 Report

The authors propose the modification in the process commonly exploited by the industry towards limiting its environmental impact, minimizing the associated production of CO2 and increasing its efficiency. The reviewer has a several comments to improve this publication:

1.      The construction of storyline can be improved. The results on total alkalinity vs. stream volume/T are part of introduction whereas the chart of production of CO2 in Inowroclaw factory is placed at the end of the results. The logical order would be the opposite.

2.      The authors should be more specific about equilibrium/non-equilibrium conditions. Some explanations should be included (e.g. the equilibrium against what etc.)?

3.      Many figures’ captions contain only symbols and cannot be understood without referring to the main text. All captions should be self-understandable (like carbonization yield (WNa) instead of just WNa).

4.      Fig. 1: The data are not sufficient to conclude about the linear trend. The additional data point at the stream volume 2 and 8 m3/h are needed to support this interpretation.

5.      Line 91: A simple scheme of the process would help a reader to follow this description.

6.      Line 278: The results from NMR and IR should be demonstrated as spectra (at least some selected if they are too many).

7.      Line 287: What calibration standards were used for IR to obtain the quantitative results?

8.      Line 289: This looks like part of the template, not the manuscript.

9.      XRD is more robust technique for the phase identification then IR especially for such complex system. At least some samples should be analyzed by XRD to confirm the obtained findings.

Author Response

(The authors gave the same response as above.)

Reviewer 3 Report

A brief summary Report:

This paper describes the modification of the soda process so as to minimize the environmental burden and minimize the production input of Na2CO3. The modifica- tions made were beneficial in reducing emissions of CO2 and increased the efficiency of the soda process, resulting in a measurable financial benefit. The obtained results are optimistic, as they assume as much as 4.87% reduction of CO2 emitted from the carbonization plant. On the scale of the plant where the experiment was carried out, this reduction of emissions of CO2 amounts to 7,93 Gg per year.

What is the temperature for to increases the return steam and temperature decreases in this process ? (page no:3/Line No.98)

How much amount of  CO2  is released to atmosphere ? (page no:18/471)

How will you reduced the CO2 by chemical reaction and what Method the author used? (page no:7/260)

What is meant by PALL RINGS ? (page no:2/85)

How did you test the  gas emission  and by which method the testing is performed? (page no:16/452)

Various types of solutions are presented in publication but few of them are implemented in industrial plant and why others are not implemented, how it differ from others (page no:2/54,55)

Calculation of non-equilibrium state it is necessary to use a specific mathematical apparatus methodology for the mathematical calculation of process parameter.  Define specific mathematical apparatus methodology ? (page no:4.141,142,143)

Author Response

Dear reviewer,
we tried to answer all questions. We have followed all comments. Detailed file attached.

Best regards

Reviewer 4 Report

Materials-1758031

Comments:

This article highlights the modification of the soda process to minimize the environmental burden and minimize the production input of Na2CO3. It is beneficial in reducing emissions of CO2 and results in a measurable financial benefit. It is an interesting topic. However, there are some comments to improve the quality of manuscript as follows.

(1)Line 15-21, In abstract section, do not present the main results and conclusions. Please add them to improve the abstract.

(2)Line102-103, Fig.1 and Fig.2, the unit should be correct into the same style. Such as [m3*h-1] and [mmol·20cm-3].

(3)Line 24-47, Line 158-159, and Line 221-227. All of abbreviations should be oblique for the letters, in particular, for the equation (7)-(14).

(4)Line 243-244, it is confusing for two “n”. such as “The identification of the suspension included the determination of alkalinity (n), total alkalinity (n),……”. They should be oblique in the context.

(5)Line 396-397, the order of six pictures is messy and disordered. Please correct.

(6)Language needs to be further improved due to the lengthy, such as “in order to……”, “Considering the fact that……”, etc.

(7)Line 320, “The equilibrium results of the carbonization process are presented in Table 1 and Fig. 2.”. Please check “Fig. 2”.

(8)Line 327. “the results are summarized in Table 2 and presented in Fig. 3.”. Please check“Fig. 3”.

(9)Line 461, and Line 470. The citations are not recommended to use in conclusions section.

(10)Please add some citations references to discuss the results deeply and to obtain better conclusions.

Author Response

(The authors gave the same response as above.)

Reviewer 5 Report

The manuscript entitled “Changes in synthetic soda ash industry and its consequence for environment” presented study using comprehensive literature review approach. The influence of different parameters was studied and analyzed. The manuscript lacks clarity and needs much improvement before further processing. It seems like this paper is directly made from thesis without putting the additional efforts required for writing manuscripts.

This reviewer recommends minor editing and resubmits for re-review.

Comments:

  • The English writing of the manuscript needs improvement. Therefore, it could benefit greatly from professional editing to improve technical writing and English.
  • Please mention your study limits and suggest some future research topics
  • In References, the sources are written in different styles. Please update the reference list.  It is necessary to bring in accordance with the requirements of the magazine for the design of References. If possible, indicate DOI.
  • The literature can be expanded by studying some of these papers.
    • Prediction of compressive strength of rice husk ash concrete through different machine learning processes
    • Compressive strength prediction of rice husk ash using multi-physics genetic expression programming
  • Please use some innovative keywords.
  • Please mention your study limits in the abstract.
  • The Conclusions should reflect what the practical application of the results obtained in this study is. In what climatic conditions should the recommendations of the authors be taken into account?
  • The authors should increase their discussion on previous related research and highlight how their study is providing a different approach or adding significantly to what has been done. The authors have to explain what is the new here in comparison with the previous studies. The novelty of the current work should be highlighted in the introduction. Please try to mention a problem that needs solving - in other words, the research question underlying your study clearer.
  • The title of the manuscript should be revised.
  • Some types of standards should be used to perform different experimental studies. Please provide details for the standards used in each study.
  • Section 4 should be discussed in detail.
  • The authors must redo the Abstract and bring it in compliance with the requirements of the journal. The scientific problem is poorly described (Background). The scientific novelty is not indicated. I recommend shortening the Abstract to 200 words. Editors strongly encourage authors to use the following style of structured abstracts, but without headings: (1) Background: Place the question addressed in a broad context and highlight the purpose of the study; (2) Methods: Briefly describe the main methods or treatments applied; (3) Results: Summarize the article's main findings; and (4) Conclusions: Indicate the main conclusions or interpretations. The abstract should be an objective representation of the article
  • It is advisable to add a flowchart at the beginning of the paper. Then the article would become more visual and structured
  • Figure 6 can be replaced with column bar chart.
  • The economic aspects are also required for sustainability in social aspect. It is suggested to authors to evaluate the cost-benefit study of this as a further investigation
  • The conclusion should be an objective summary of the most important findings in response to the specific research question or hypothesis. A good conclusion states the principal topic, key arguments and counterpoint, and might suggest future research. It is important to understand the methodological robustness of your study design and report your findings accordingly. Please improve your conclusion section.

Author Response

(The authors gave the same response as above.)

Round 2

Reviewer 2 Report

The authors have addressed the reviewer's comments sufficiently.

Reviewer 5 Report

Accept